# Characterization of the Prion Protein Binding Properties of Antisense Oligonucleotides

**DOI:** 10.3390/biom10010001

**Published:** 2019-12-18

**Authors:** Andrew G. Reidenbach, Eric Vallabh Minikel, Hien T. Zhao, Stacy G. Guzman, Alison J. Leed, Michael F. Mesleh, Holly B. Kordasiewicz, Stuart L. Schreiber, Sonia M. Vallabh

**Affiliations:** 1Chemical Biology and Therapeutics Science Program, Broad Institute of MIT and Harvard, Cambridge, MA 02142, USA; areidenb@broadinstitute.org (A.G.R.); eminikel@broadinstitute.org (E.V.M.); stacy.guzman@pennmedicine.upenn.edu (S.G.G.); stuart_schreiber@harvard.edu (S.L.S.); 2Stanley Center for Psychiatric Research, Broad Institute of MIT and Harvard, Cambridge, MA 02142, USA; 3Prion Alliance, Cambridge, MA 02139, USA; 4Ionis Pharmaceuticals, Carlsbad, CA 92010, USA; hzhao@ionisph.com (H.T.Z.); hkordasiewicz@ionisph.com (H.B.K.); 5Exceptional Research Opportunities Program (EXROP), Howard Hughes Medical Institute (HHMI), Chevy Chase, MD 20815, USA; 6Perelman School of Medicine, University of Pennsylvania, Philadelphia, PA 19104, USA; 7Center for the Development of Therapeutics, Broad Institute of MIT and Harvard, Cambridge, MA 02142, USA; aleed@broadinstitute.org (A.J.L.); mmesleh@broadinstitute.org (M.F.M.); 8Department of Chemistry & Chemical Biology, Harvard University, Cambridge, MA 02138, USA

**Keywords:** prion, antisense oligonucleotide, biomarker, cerebrospinal fluid, neurodegeneration, isothermal titration calorimetry, dynamic light scattering, nuclear magnetic resonance

## Abstract

Antisense oligonucleotides (ASOs) designed to lower prion protein (PrP) expression in the brain through RNase H1-mediated degradation of PrP RNA are in development as prion disease therapeutics. ASOs were previously reported to sequence-independently interact with PrP and inhibit prion accumulation in cell culture, yet *in vivo* studies using a new generation of ASOs found that only PrP-lowering sequences were effective at extending survival. Cerebrospinal fluid (CSF) PrP has been proposed as a pharmacodynamic biomarker for trials of such ASOs, but is only interpretable if PrP lowering is indeed the relevant mechanism of action in vivo and if measurement of PrP is unconfounded by any PrP–ASO interaction. Here, we examine the PrP-binding and antiprion properties of ASOs in vitro and in cell culture. Binding parameters determined by isothermal titration calorimetry were similar across all ASOs tested, indicating that ASOs of various chemistries bind full-length recombinant PrP with low- to mid-nanomolar affinity in a sequence-independent manner. Nuclear magnetic resonance, dynamic light scattering, and visual inspection of ASO–PrP mixtures suggested, however, that this interaction is characterized by the formation of large aggregates, a conclusion further supported by the salt dependence of the affinity measured by isothermal titration calorimetry. Sequence-independent inhibition of prion accumulation in cell culture was observed. The inefficacy of non-PrP-lowering ASOs against prion disease in vivo may be because their apparent activity in vitro is an artifact of aggregation, or because the concentration of ASOs in relevant compartments within the central nervous system (CNS) quickly drops below the effective concentration for sequence-independent antiprion activity after bolus dosing into CSF. Measurements of PrP concentration in human CSF were not impacted by the addition of ASO. These findings support the further development of PrP-lowering ASOs and of CSF PrP as a pharmacodynamic biomarker.

## 1. Introduction

Prion disease is a fatal, incurable neurodegenerative disease caused by misfolding of the prion protein (PrP) [1]. Antisense oligonucleotides (ASOs) now in preclinical development for prion disease aim to lower PrP expression in the brain, a therapeutic strategy supported by strong genetic proof of concept [2]. These ASOs are being designed to trigger RNase H1 cleavage of PrP mRNA, a well-established mechanism of action of ASOs [3,4,5]. Early studies of ASOs for prion disease, however, left doubt as to whether their efficacy was mediated by RNase H or by an aptameric mechanism [6]. PrP interacts with diverse polyanions [7,8] and nucleic acids [9]. Indeed, oligonucleotides with the phosphorothioate (PS) backbone widely used in ASOs [10] have exhibited sequence-independent antiprion activity [6,11,12]. PS ASOs have been reported to bind PrP in vitro with nanomolar affinity [11] in a sequence-independent manner, lower PrP expression in cultured cells with micromolar potency [6,12], and even strongly inhibit the accumulation of misfolded PrP in prion-infected cultured cells, with low nanomolar potency [6,11,12]. In recent studies, PrP-lowering ASOs delivered by intracerebroventricular bolus injection potently extended survival of prion-infected mice, whereas a non-PrP-lowering control ASO conferred no survival benefit [13]. These findings establish RNA-mediated PrP lowering as the mechanism of action of ASOs against prion disease in vivo. Importantly, this mechanism lends itself to measurement of cerebrospinal fluid (CSF) PrP concentration as a pharmacodynamic biomarker for ASO activity [14,15,16]. But the question remains as to why the apparently potent interactions between ASOs and PrP appear not to contribute to in vivo efficacy. Here, we revisit the interaction between ASOs and PrP in vitro and in cell culture. We replicate previous reports of a sequence-independent interaction, but provide evidence that this interaction involves aggregation, and that it requires ASO concentrations that may not be sustained in bolus dosing paradigms. Either or both of these factors may explain the lack of sequence-independent activity in vivo.

## 2. Materials and Methods

### 2.1. Recombinant Protein Preparation

Recombinant full-length human PrP (HuPrP23-230) was expressed in *Escherichia coli* and purified from inclusion bodies by fractionation, denaturation, and refolding on a Ni-NTA column as described [17]. The vector was a generous gift from Byron Caughey’s laboratory at NIAID Rocky Mountain Labs.

### 2.2. Test Compounds

ASOs were prepared and purified as described [13]. Heparin was purchased as a sodium salt purified from porcine intestinal mucosa (Sigma H3393-50KU). This natural source of heparin is of heterogeneous molecular weight, so it is considered in mass/vol rather than molarity terms throughout this study.

### 2.3. Isothermal Titration Calorimetry

ITC was performed on a MicroCal Auto-ITC200 instrument (Malvern Panalytical, Inc, Westborough, MA, USA). HuPrP23-230 and ASOs were prepared in dialysis-matched (and, where applicable, NaCl-matched) buffer containing 20 mM phosphate at pH 7.0. Protein was placed in the cell and compound in a syringe, and each experiment was paired with a control experiment injecting the compound into dialysis buffer. Cell temperature was set to 25 °C. Each experiment involved 40 injections of 2 s duration (1 µL total volume), at 120 s spacing, with a 5 s filter period, 10 µcal/s reference power, 750 rpm stirring speed, and high gain feedback mode. Data were processed, background-subtracted and curves fit in “single set of sites” mode in the manufacturer’s Origin software to obtain thermodynamic parameters.

### 2.4. Nuclear Magnetic Resonance

The ^15^N-labeled recombinant HuPrP23-230 was prepared by growing the *E. coli* in ^15^N-enriched cell growth medium (Cambridge Isotope Laboratories CGM-1000-N, Tewksbury, MA, USA) supplemented with Overnight Express Autoinduction Medium (Millipore-Sigma 71300-4, Burlington, MA, USA). ^1^H-^15^N TROSY spectra were acquired on a 600 MHz Bruker Avance III spectrometer (Billerica, MA, USA) equipped with a 5 mm QCI cryoprobe. Samples contained 50 μM HuPrP23-230 in 20 mM sodium phosphate, pH 7.0, with 90% H_2_O/10% D_2_O.

### 2.5. Dynamic Light Scattering

HuPrP23-230 and active ASO 1 hydrodynamic radii (Rh) were measured using a DynaPro Plate Reader II (Wyatt Technology, Santa Barbara, CA, USA) with five 3 s acquisitions. HuPrP23-230 and ASO were used at 50 μM in a 20 mM sodium phosphate pH 7.0 buffer, and all measurements were performed in a 384-well format (25 μL sample/well) at 25 °C. Regularization analysis was used for curve fitting, and Rayleigh spheres were used as the model to determine the Rh.

### 2.6. Tissue Culture

ScN2a cells (mouse neuroblastoma cells chronically infected with the RML strain of prions) [18], a generous gift from Sina Ghaemmaghami, were cultured in DMEM media (Life Technologies 11965, Carlsbad, CA, USA) supplied with 10% FBS, 1% glutamax, and 1% pen/strep. For RNA measurements, 20,000 cells were seeded per well in 96-well plates and indicated ASO concentrations were applied in culture media. For protein measurements, 50,000 cells were seeded per well in 12-well plates and indicated ASO concentrations were applied in culture media.

### 2.7. qPCR

Cells were lysed in 300 µL of RLT buffer (Qiagen, Valencia, CA, USA) containing 1% (*v*/*v*) 2-mercaptoethanol (BME, Sigma Aldrich, St. Louis, MO, USA). RNA was isolated with an RNeasy 96 Kit (Qiagen) that included in-column DNA digestion with 50 U of DNAse 1 (Thermo Fisher Scientific 18068015, Waltham, MA, USA). *Prnp* mRNA in cells was quantified as previously described [13,19]. Briefly, qPCR was performed on a StepOne Realtime PCR system (Applied Biosystems, Foster City, CA, USA), with results normalized first to the housekeeping gene cyclophilin A (*Ppia*) and then to the level in saline-treated cells. Primers and probes were as follows: *Prnp* forward primer TCAGTCATCATGGCGAACCTT, reverse primer AGGCCGACATCAGTCCACAT, and probe CTACTGGCTGCTGGCCCTCTTTGTGACX; *Ppia* forward primer TCGCCGCTTGCTGCA, reverse primer ATCGGCCGTGATGTCGA, and probe CCATGGTCAACCCCACCGTGTTCX.

### 2.8. Immunoblotting and Proteinase K Digest

Cells were lysed at 72 h with lysis buffer containing 0.5% Triton X-100 and 0.5% DOC in PBS and spun at 3000× *g* for 5 min at 4 °C. Supernatants were subjected to BCA (Thermo Fisher Scientific 23227) to determine protein concentration. Part of the supernatants were saved as PK-sensitive material for total PrP. For PK-resistant PrP^Sc^, 25 µg total protein was subjected to PK digestion (P6556) using 1 µg/mL PK for 5 min. Digestion was stopped by adding 1 µL of 25 mM Pefabloc. Then, 20 µg total protein for each sample was run on 4–12% Bis-Tris gels (Thermo Fisher Scientific NP0322) with MES buffer (Thermo Fisher Scientific NP002), probed with 1:500 6D11 antibody (Biolegend 808001) in 5% milk TBS with 0.1% Tween overnight, detected with IRDye 680RD goat-anti-mouse IgG, imaged, and quantified using LI-COR Odyssey.

### 2.9. Enzyme-linked immunosorbent assay (ELISA)

Studies on human samples were approved by the Broad Institute’s Office of Research Subjects Protection (ORSP-3587). Experiments used previously reported [14] human CSF samples from normal pressure hydrocephalus patients, provided by the Mind Tissue Bank at Massachusetts General Hospital. CSF was stored at −80 °C until use. To minimize PrP loss to plastic, experimental CSF handling was minimized. ASO and/or heparin stocks were prepared in PBS in series at 10× the desired final concentration. These stocks were then spiked into separate aliquots of the same CSF sample, premade either with 0.03% CHAPS or with no additive as indicated. After gentle mixing, spiked CSF samples were incubated at room temperature for one hour prior to being assayed. PrP was quantified using the BetaPrion Human ELISA kit (Analytik-Jena) according to the manufacturer’s instructions.

### 2.10. Data Analysis

Data were analyzed in R 3.5.1. Dose-response curves were fit using the drc package [20] with a four-parameter log logistic curve fit (LL.4). Raw data and source code sufficient to reproduce the figures herein are available online at https://github.com/ericminikel/aso_in_vitro.

## 3. Results

We used isothermal titration calorimetry (ITC) to estimate the thermodynamic parameters of interactions between full-length recombinant human prion protein (HuPrP23-230) and eight previously described [2,6,13] ASOs (Table 1) as well as heparin, a polyanionic positive control [8] (Figure 1). All eight ASOs and heparin showed isotherms consistent with a saturable binding event driven by enthalpy, despite a decrease in entropy (Figure 1C–E). Calculated affinity (*K*_d_) values were similar, in the low- to mid-nanomolar range (Figure 1C), for all tested ASOs including a fully phosphorothioated 20 mer used as a previous generation of control ASO [6] and the mixed phosphorothioate/phosphodiester (PS/PO) backbone [21] ASOs with MOE and/or cEt modifications employed more recently [13]. This suggested that the lack of in vivo efficacy of non-PrP-targeting ASOs in recent studies was not due to chemical differences in PrP affinity between ASO chemistries.

In an effort to characterize the binding site of ASOs on PrP, we performed protein-observed ^1^H–^15^N transverse relaxation-optimized spectroscopy [22] (TROSY) nuclear magnetic resonance (NMR) on 50 µM ^15^N-labeled HuPrP23-230, with or without 50 µM ASO. While the control PrP spectrum appeared as expected, virtually all peaks representing PrP backbone amides disappeared when ASO was added (Figure 2A). Inspection of the NMR tubes after the fact revealed the presence of an opaque white solid, suggesting PrP had precipitated (Figure 2B). Subsequent experiments confirmed the formation of visible aggregates within seconds or minutes of addition of ASO to PrP. To quantify this aggregation, we performed dynamic light scattering (DLS) on ASO, PrP, or mixtures of the two. Mass histograms obtained by DLS showed a ~1 nm hydrodynamic radius for 50 µM ASO alone and ~2 nm for 50 µM PrP alone, but when the two were mixed, both of these peaks vanished, replaced with a single peak with a mean hydrodynamic radius of ≥30 nm. These findings suggested that ASOs can aggregate together with recombinant PrP when the two are mixed in vitro.

The foregoing results suggested that the PrP–ASO interaction observed in vitro might arise from aggregation. As dependence upon ionic strength has been observed for other aggregators [23], and given the net positive charge (+7) of PrP and negative charge of ASOs, we then tested the affinity of the PrP–ASO interaction by ITC at variable salt concentrations. For both ASOs tested (Figure 3A,B), higher salt concentration corresponded to a reduced change in enthalpy and a lower binding curve slope, both indicative of weaker affinity. At supraphysiological salt levels, 500 mM, no binding was observed.

Given that the biophysical parameters of the ASO–PrP interaction appeared similar across ASOs, we sought to replicate, with ASOs recently tested in vivo [13], the previous reports of sequence-independent lowering of misfolded PrP accumulation in cell culture. Chronically RML prion-infected mouse neuroblastoma (ScN2a) cells [18] were exposed to ASOs in a gymnotic system (without transfection or electroporation) and harvested for PrP mRNA quantification by qPCR after 24 h, or for PrP quantification with or without proteinase K digestion after 72 h. Substantial reduction of PrP mRNA and total PrP were observed only for PrP-lowering ASOs and only at micromolar doses (Figure 4A,B), consistent with limited productive uptake in ScN2a cells. These effective concentrations are orders of magnitude above the EC_50_ values observed for RNase H1-mediated activity of potent ASOs either after transfection or in cells exhibiting productive free uptake of ASOs [24,25]. In contrast, both PrP-lowering ASOs and a non-targeting control ASO antagonized proteinase K (PK)-resistant PrP^Sc^, consistent with previous reports, albeit with substantial inhibition only at near-micromolar concentrations (Figure 4C). PrP^Sc^ has a longer half-life than PrP^C^, and its clearance by oligonucleotides is time-dependent [6], so the higher effective concentration observed here may reflect the shorter treatment period (3 days) compared to that previously used (7 days).

Measurement of PrP concentration in CSF has been proposed as a pharmacodynamic biomarker for clinical trials of PrP-lowering ASOs [14,16], meaning, it will be important to be able to accurately measure CSF PrP in the presence of ASO. We sought to determine whether the addition of ASOs would confound measurement of PrP in CSF. We tested a range of ASO concentrations from 1 ng/mL (the lower limit of quantification for some ASOs in CSF [26]) to 1 mg/mL (the highest concentration that might be reached immediately after bolus dosing in a human [27]). We began with CSF containing 0.03% CHAPS, as this helps to minimize pre-analytical variability [14,16]. Addition of ASO up to 1 mg/mL did not result in any detectable change in CSF PrP concentration measured by ELISA (Figure 5A). The same was true for heparin (Figure 5B). Even in neat CSF lacking detergent, addition of ASO had no effect on the detected PrP concentration (Figure 5C).

## 4. Discussion

Phosphorothioate oligonucleotides have been reported to bind PrP in vitro in a sequence-independent manner with nanomolar affinity and to antagonize misfolded PrP accumulation in cell culture with nanomolar efficacy [6,11,12], yet in vivo, we have reported that only antisense sequences targeting the PrP RNA are effective at extending survival in prion-infected mice [13]. To reconcile these observations, here, we revisited the binding of ASOs and PrP. By ITC, using recombinant PrP and by immunoblotting in prion-infected ScN2a cells, we replicated the sequence-independent in vitro binding and cell culture antiprion potency of ASOs, and showed that these properties are maintained in ASOs that incorporate mixed PS/PO backbones with MOE and/or cEt sugar modifications.

However, we also found evidence, by NMR, DLS, and observation by the naked eye, that the in vitro interaction between otherwise monomeric ASOs and monomeric PrP involves the formation of large aggregates, apparently containing both the ASO and PrP. Although we have not deeply characterized these aggregates to determine whether they are protease-resistant and/or represent liquid–liquid phase separation [28,29], our general findings are in line with several reports describing PrP aggregation in the presence of nucleic acids [28,30,31,32,33]. Compounds that aggregate are often considered “pan-assay interference compounds” (PAINS) [23,34,35]. The inhibitory behavior of such compounds is often dependent upon ionic strength [23], as we observe here for interactions between ASOs and PrP. The inherent limitations of DLS preclude us from determining whether this aggregation event occurs at lower concentrations of ASO and PrP, such as the nanomolar concentrations used in previous in vitro binding studies [11], or in complex mixtures such as cell culture media [6,11,12]. Thus, our results urge caution around the interpretation of ASO–PrP binding studies, but do not prove that findings in previous reports were necessarily the result of aggregation.

In a small-molecule drug-discovery campaign, a compound found to exhibit or trigger aggregation would generally be labeled a false positive hit and removed from consideration. The fact that ASOs do exhibit sequence-independent antiprion activity in cell culture argues that the binding event may not be purely an in vitro artifact, but might correspond to some genuine interaction at the surface of cultured cells, in line with previous reports [11,32]. Nevertheless, ASOs have over 60 documented interacting proteins [36], PrP binds many polyanions [7], and indeed, many other polyanions non-specifically bind a large number of different proteins [37], so the ASO–PrP interaction may not be at all unique or specific. Moreover, several other compounds now understood to be PAINS, such as curcumin and epigallocatechin gallate [35,38], were also shown to reduce misfolded PrP accumulation in cell culture [39,40] and yet lacked any clear in vivo efficacy against prion infections of the central nervous system (CNS) [39,41,42]. It is therefore perhaps not surprising that a non-PrP-lowering ASO might exhibit a similar profile in vitro and in cell culture, and yet lack activity in vivo.

Alternatively, even if ASO–PrP binding as characterized here does occur in vivo, our characterization of the biophysical parameters of this interaction suggests that pharmacokinetics might also be sufficient to explain why non-PrP-lowering ASOs do not extend survival in prion-infected mice. Following bolus injection into CSF, ASOs are rapidly either absorbed into CNS tissue or cleared into plasma [43]. In clinical trials of the ASO nusinersen for spinal muscular atrophy, drug concentration dropped to only ~3 ng/mL (0.4 nM) in CSF by 7 days post-dose [26]. Thus, in a periodic bolus dosing paradigm, concentrations of ASO in the extracellular space and/or in relevant endosomal/lysosomal compartments for PrP binding might not be maintained above the low- to mid-nanomolar *K*_d_ values for PrP binding for more than a few hours or days. In contrast, cytosolic and nuclear concentrations of ASO remain above the effective concentration for RNase H1-mediated activity for months following bolus dosing [2].

Importantly, despite the interaction between ASO and PrP and despite the ability of ASOs to trigger recombinant PrP precipitation, we find that the presence of ASO in CSF, even at concentrations well above those likely to be sustained in human dosing, does not interfere with quantification of endogenous PrP by ELISA. The fact that ELISA reactivity of CSF PrP is apparently unaffected by the presence of ASO may be due in part to the low concentration of PrP in CSF, the presence of other proteins in CSF, and the presence of protein and detergent in the blocking buffer used for ELISA. Measurement of CSF PrP should be able to report on the PrP-lowering effects of an ASO in the human CNS in clinical trials.

## 5. Conclusions

Our data support the further development of ASOs for prion disease on the basis of RNase H1-mediated lowering of PrP RNA as a mechanism of action and the measurement of PrP in CSF as a pharmacodynamic biomarker to monitor this effect.

## Figures and Tables

**Figure 1 biomolecules-10-00001-f001:**
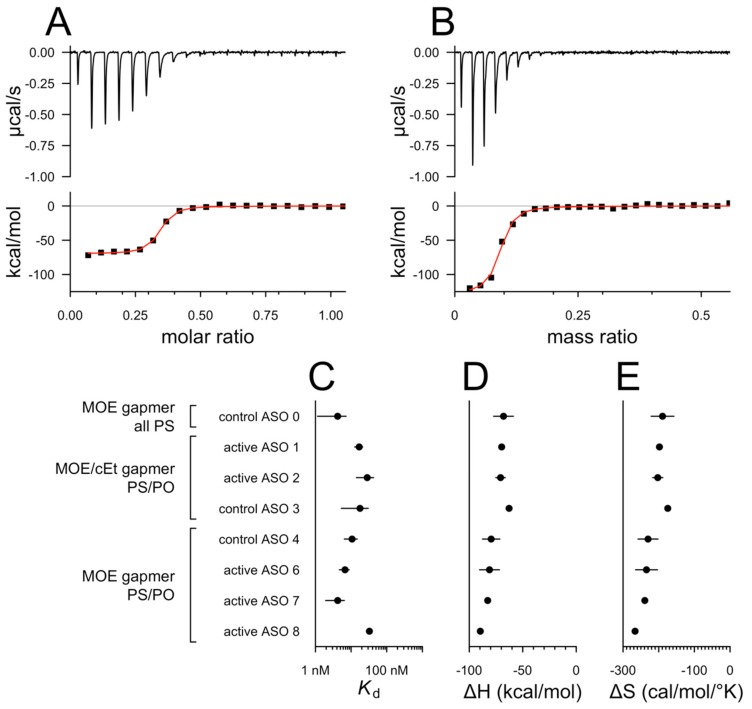
Isothermal titration calorimetry. Isotherms for (**A**) active ASO 1 and (**B**) heparin, a positive control. These isotherms were consistent with half of the protein being bound at 21 nM or 125 µg/L ASO 1 and at 518 µg/L heparin. Because heparin is of heterogeneous molecular weight, results are presented in terms of mass stoichiometry to prion protein (PrP); a molar *K*_d_ was not calculated. Thermodynamic parameters calculated from ITC for various ASOs: (**C**) *K*_d_, (**D**) enthalpy, and (**E**) entropy.

**Figure 2 biomolecules-10-00001-f002:**
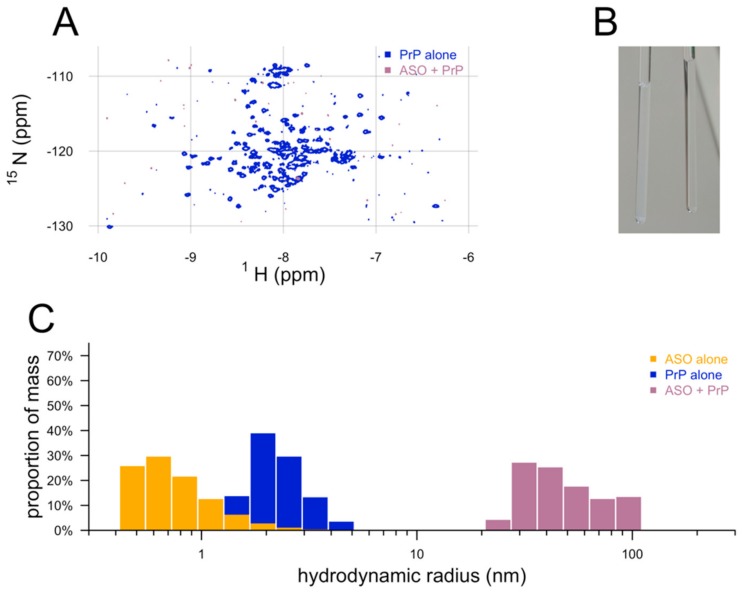
Aggregation of ASOs and recombinant PrP observed by NMR, DLS, and the naked eye. (**A**) 2D TROSY NMR spectrum of ^15^N-labeled HuPrP23-230 with and without ASO. (**B**) Photograph of PrP/ASO mixture (left) and PrP alone (right) from panel (A) in the NMR tubes. (**C**) Mass histograms from dynamic light scattering of ASO, PrP, and ASO + PrP.

**Figure 3 biomolecules-10-00001-f003:**
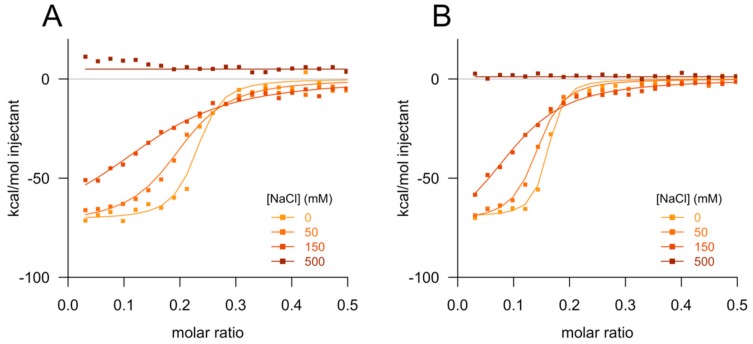
ASO–PrP interaction depends upon ionic strength. Isothermal titration calorimetry fitted curves for (**A**) active ASO 1 and (**B**) control ASO 4 versus salt concentration.

**Figure 4 biomolecules-10-00001-f004:**
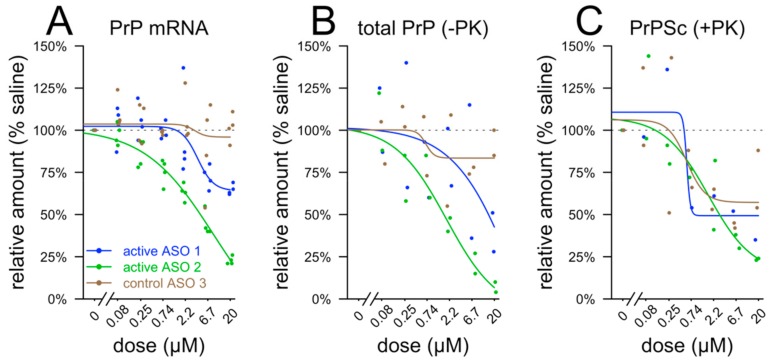
Dose-dependent antiprion activity of ASOs in ScN2a cells. All experiments relied on free uptake of ASOs without transfection or electroporation. Note the log x-axis in all panels. (**A**) PrP mRNA quantified by qPCR after 24 h exposure, (**B**) total PrP quantified after 72 h exposure by Western blot without proteinase K (PK) digestion, and (**C**) protease-resistant PrP^Sc^ quantified after 72 h exposure by immunoblot after PK digestion. Dose response was determined by four parameter log logistic curves (see Methods). Western blot images for panels B and C are available in this study’s online data repository (see Section 2.10.).

**Figure 5 biomolecules-10-00001-f005:**
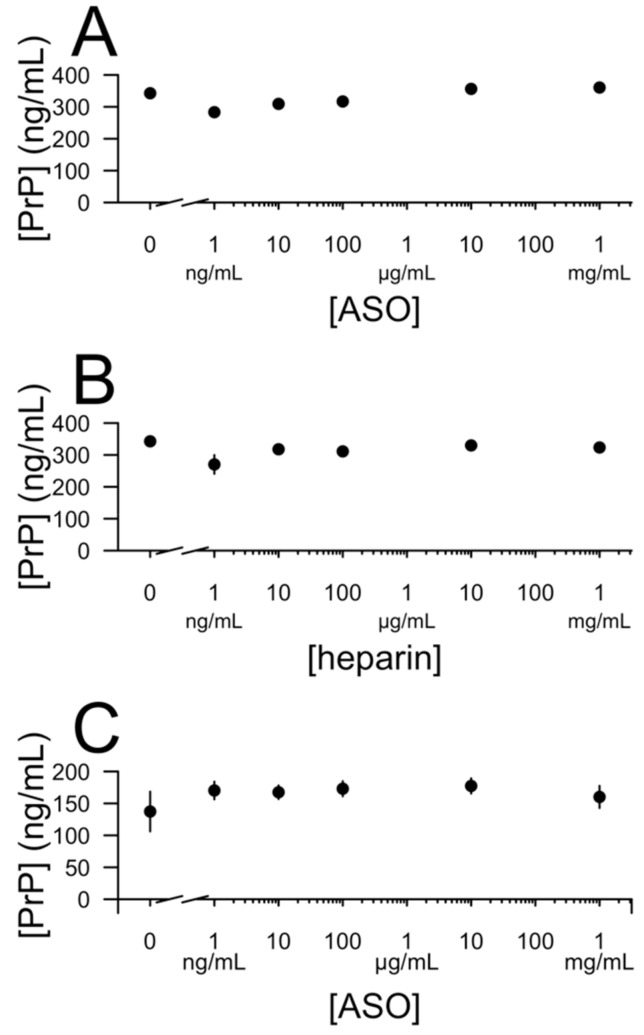
Presence of ASO does not affect quantification of PrP in cerebrospinal fluid (CSF). (**A**) Active ASO 6 spiked into CSF with 0.03% CHAPS. (**B**) Heparin spiked into CSF with 0.03% CHAPS. (**C**) Active ASO 6 spiked into neat CSF.

**Table 1 biomolecules-10-00001-t001:** Previously described [10] antisense oligonucleotide (ASO) chemistries are indicated by the following color and lettering scheme. Black: unmodified deoxyribose (2′H); orange: 2′ methoxyethyl (MOE); blue: 2′-4′ constrained ethyl (cET). Unmarked linkages: phosphorothioate (PS); linkages marked with o: normal phosphodiester (PO); mC: 5-methylcytosine. Control ASO 0 is the compound referred to as ASO 923 in a previous report [6]. MW, molecular weight; nt, nucleotides (length).

Treatment	Annotated Sequence	nt	Chemistry	MW (kDa)
control ASO 0	mCmCTTmCmCmCTGAAGGTTmCmCTmCmC	20	PS MOE	7.15
active ASO 1	mCT o A o T TTAATGTmC A o G o TmCT	17	PS/PO MOE/cEt	5.99
active ASO 2	TT o G o mC AATTmCTAT mC o mC o AAA	17	PS/PO MOE/cEt	5.98
control ASO 3	mCG o mC o T TATAmCTAA T mC o A o TAT	17	PS/PO MOE/cEt	5.98
control ASO 4	mCmCoToAoTAGGAmCTATmCmCAoGoGoAA	20	PS/PO MOE	7.13
active ASO 6	mCToTomCoTATTTAATGTmCAoGoTmCT	20	PS/PO MOE	7.07
active ASO 7	TAoGomComCTTTGTACCTTAoAomCmCA	20	PS/PO MOE	7.08
active ASO 8	GmComCoAAGGTTmCGmCmCoAoTGA	17	PS/PO MOE	6.12

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
