# Peer review of "Characterization of the Prion Protein Binding Properties of Antisense Oligonucleotides"

_biomolecules, 2019, doi:10.3390/biom10010001_

Round 1

Reviewer 1 Report

Dear Editor and authors,

The work from Reidenbach and collaborators correlate the antiprion properties of antisense oligonucleotides (ASOs) designed to lower prion protein (PrP) levels in the brain with PrP binding affinity and whether ASOs will interfere with detection of PrP in the cerebrospinal fluid. The work is well-written and the experimental approach is well designed. The authors employ a myriad of techniques to dissect interaction of PrP with the ASOs and the effects derived from this interaction. I recommend publication in Biomolecules after minor changes.

General comments:

Aggregation of PrP induced by polyanions has long been investigated; and in particular, our work regarding interaction with DNA, RNA and glycosaminoglycans show similar effects of those here reported (such as NMR, particle size by DLS, ITC) (PMID: 28513534; PMID: 22691027; PMID: 18456654; PMID: 16866364; PMID: 11604397. In addition, we have a recent paper that can provide information to your work where we show that this transient aggregation caused by interaction with nucleic acids is mediated by liquid-liquid phase separation processes (preprint: http://biorxiv.org/cgi/content/short/659037v1 and FASEB J: https://onlinelibrary.wiley.com/doi/abs/10.1096/fj.201901897R). In my opinion, NA induced PrP aggregation can preclude the use of such molecules in therapy. However, searching for PrP stabilizing and anti-aggregation ASOs is an interesting approach.

Best regards,

Minor comments:

Reference 2 is a dissertation; I do not know if the journal accepts this kind of citation;

Introduction: The authors might consider referring to the work of Silva, JL and Cordeiro, Y., that characterized PrP aggregation induced by nucleic acids.

Author Response

We thank the reviewer for this thorough read and thoughtful comments on our manuscript and for pointing us to these highly relevant references. We have added citations to these papers in our introduction (“PrP interacts with diverse polyanions [7,8], and nucleic acids  [9]”) and two places in the discussion (“Although we have not deeply characterized these aggregates to determine whether they are protease-resistant and/or represent liquid-liquid phase separation [27,28], our general findings are in line with several reports describing PrP aggregation in the presence of nucleic acids [27,29–32].” and “…might correspond to some genuine interaction at the surface of cultured cells, in line with previous reports [11,29]”). We have not performed experiments to rule out the existence of LLPS droplets in our experiments, as we believe this would be out of scope for the present manuscript, but we agree it is a possibility and we hope that the incorporation of this possibility into the discussion adequately addresses this concern.

We believe that PrP-lowering ASOs are a promising therapeutic modality for prion disease, as our in vivo data demonstrate a significant survival benefit for PrP-lowering ASOs but do not demonstrate any sequence-independent effect.

Reviewer 2 Report

This paper systematically characterizes the interaction of antisense oligonucleotides (ASOs) with the prion protein (PrP). Using isothermal titration calorimetry, the authors show that ASOs bind to recombinant PrP in a sequence independent manner, with Kds in the nanomolar range. By titrating the salt concentration, they are able to show that this is likely an ionic interaction between positively charged PrP (+7) and negatively charged ASOs. Binding of ASOs to recombinant PrP results in the formation of aggregates and the precipitation of PrP from solution. Using ScN2a cells, the authors show that while only active ASOs reduce PrP mRNA levels, both active and control ASOs lower PrPSc levels. Notably, the authors show that ASOs do not affect the measurement of PrP levels in CSF, which is important since CSF PrP levels are being proposed for use as a biomarker in future ASO trials for human prion disease.

This is an interesting, well-written paper that will be of particular interest to those scientists working on developing ASOs for the treatment of neurodegenerative diseases. The methodology and conclusions are sound. The only thing missing is an investigation into whether ASOs induce PrP aggregation in cells (see comment #1). Overall, I have a favorable impression of this manuscript.

Comments:

The idea that ASO-dependent sequestration of PrPC in aggregates (in a sequence-independent fashion) prevents its conversion to PrPSc makes a lot of sense. However, there is a bit of a knowledge gap between the recombinant PrP experiments and the ScN2a experiments. Do the authors know whether ASOs cause PrPC to form aggregates in actual cells? This is not a trivial question, since recombinant PrP lacks N-glycans, which could conceivably influence the binding of ASOs to PrP. This could easily be addressed by adding ASOs to uninfected N2a cells and then monitoring any changes in the solubility of endogenous PrPC. It would also be worthwhile to see if a percentage of PrP is found to be insoluble in the CSF as a result of ASO treatment. Figure 2 nicely shows that addition of ASOs to recombinant PrP results in the formation of some sort of aggregated material. Given the importance of PrP aggregates to prion biology, the authors should perform some quick biochemical analyses to see if these aggregates exhibit any of the properties of prions. For instance, are they resistant to digestion with proteinase K? Have the authors considered that these “aggregates” may not actually represent true aggregates but liquid-liquid phase separation of PrP, as has been reported by Kostylev et al., (“Liquid and Hydrogel Phase of PrPC Linked to Conformational Shifts and Triggered by Alzheimer’s Amyloid-beta Oligomers”, Molecular Cell, 2018)? In Figure 4, the authors should show a representative Western blot (at least for PK-digested PrP) for each of the different ASOs.

Author Response

We would thank the reviewer for these detailed comments. We agree that it is still not clear how ASOs affect PrPC clustering or aggregation in cell culture, and we have not characterized the biochemical nature of the aggregates. While additional experiments on these subjects would be out of scope for the present manuscript, we have addressed this limitation in the Discussion and have added the indicated citation: “Although we have not deeply characterized these aggregates to determine whether they are protease-resistant and/or represent liquid-liquid phase separation [27,28], our general findings are in line with several reports describing PrP aggregation in the presence of nucleic acids [27,29–32].”

We have uploaded the original Western blots for Figure 4 into the public data repository for this study: https://github.com/ericminikel/aso_in_vitro/

Reviewer 3 Report

The manuscript by Reidenbach et al revisits the discrepancy between the efficacy of non-prion targeted ASOs in vitro vs in vivo compared to prion-targeted ASOs. They further demonstrate that direct addition of ASOs into CSF does not affect the apparent prion protein concentration detected in the CSF. They therefore proposed CSF PrP levels as biomarker for clinical trials using PrP-lowering ASOs.

The manuscript is well written and provides an in vitro, biophysical justification beyond recent animal studies to pursue the development of PrP-lowering ASOs as a treatment for prion diseases. There are some points, however, that should be discussed:

The techniques used in this study should allow for the calculation of stoichiometry. If this is indeed a 1:1 stoichiometry as the isotherms suggest, then precipitation likely occurs at a 1:1 mixture, as indicated by the DLS. This would then imply that, even though not observable at the low concentrations of PrP in the CSF, that precipitation is in fact occurring, even at the low end of the ASO concentrations. This precipitation could account for both some of the toxicity caused by the ASOs in previous studies, particularly at the bolus doses, and also the efficacy of the PrP-targeted ASOs at the low to sub-stoichiometric doses that are maintained. This precipitation may not result in a noticeable concentration drop by ELISA as the epitopes may still be available. Furthermore, this effect may be exacerbated later in disease as prion aggregates are already forming and could have a more pronounced precipitation effect. These points should be considered and discussed further.

Author Response

We thank the reviewer for these comments and critical reading of our manuscript.

The manuscript does in fact provide some, albeit limited, data on stoichiometry: in the isotherm in Figure 1a, saturation is observed at a ~1:3 ASO:PrP molar ratio. This implies that ASO should be the limiting reagent, or in other words, 1 mg/mL ASO added to CSF (Figure 5) should be more than enough to trigger precipitation of all available PrP. We have added a section to discussion acknowledging the several possible expanations for what we observed: “The fact that ELISA reactivity of CSF PrP is apparently unaffected by the presence of ASO may be due in part to the low concentration of PrP in CSF, presence of other proteins in CSF, and the presence of protein and detergent in the blocking buffer used for ELISA.”

As for the suggestion that precipitation could be related to ASO toxicity, we note that no differences in binding parameters were observed among the ASOs studied here (Figure 1), which include ASOs that were well-tolerated and ASOs that were not well-tolerated at a neuropathological disease stage where PrP aggregates have already formed (Raymond et al, PMID: 31361599).